# Observation of the spiral spin liquid in a triangular-lattice material

N. D. Andriushin [1] ✉, S. E. Nikitin [2], Ø. S. Fjellvåg [2,3], J. S. White [2], A. Podlesnyak [4], D. S. Inosov [1,5], M. C. Rahn [1,6], M. Schmidt[7], M. Baenitz [7] ✉ & A. S. Sukhanov[1,6] ✉

The spiral spin liquid (SSL) is a highly degenerate state characterized by a continuous contour or surface in reciprocal space spanned by a spiral propagation vector. Although the SSL state has been predicted in a number of various theoretical models, very few materials are so far experimentally identified to host such a state. Via combined single-crystal wide-angle and small-angle neutron scattering, we report observation of the SSL in the quasi-two-dimensional delafossite-like $AgCrSe_2$. We show that it is a very close realization of the ideal Heisenberg $J_1$–$J_2$–$J_3$ frustrated model on the triangular lattice. By supplementing our experimental results with microscopic spin-dynamics simulations, we demonstrate how such exotic magnetic states are driven by thermal fluctuations and exchange frustration.

The phase transition on cooling from a paramagnetic to a magnetically ordered state necessarily breaks certain symmetries of the system. The static nature of magnetic order implies breaking of the time-reversal symmetry, and most often some of the rotational and/or translational symmetries are also lost. Such phase transitions take place in the vast majority of magnetic materials with non-negligible interatomic magnetic interactions. Defects and disorder-free systems of interacting spins that do not exhibit long-range order down to zero temperature are called spin liquids, which seem to be exceptionally rare in nature[1–3]. Spin liquids are recognized to demonstrate unconventional behavior and exotic quasi-particle excitations[4–6] stemming from their extensive degeneracy of the ground state. The magnetic order in spin liquids is precluded by strong frustration. Instead, they feature a strongly-correlated state that still preserves the spin rotational symmetries of the paramagnetic system down to zero temperature. The spin-liquid state is often realized on a phase boundary between two ordered states, described in terms of propagation vectors anchored to the high-symmetry points of the Brillouin zone (BZ) boundary. For instance, the theoretical studies of the quantum $J_1$–$J_2$ Heisenberg

model on the triangular lattice[7–9] revealed that the quantum spin liquid appears for a narrow parameter range between the 120° and stripe orders. However, recent works also showed that a highly-fluctuating state, akin to the quantum spin liquid, can also be formed in the classic regime at an arbitrary non-zero wavenumber. The latter was termed the spiral spin liquid (SSL) as the spins in such a state maintain a well-defined spiral pitch[10–16].

Hence, the SSL is a classical spin liquid state, which lacks a long-range order but exhibits strong short-range correlations that retain the periodicity of the spin spiral. Unlike an ordinary spin spiral state, the propagation vector of the SSL spans a contour or a surface in reciprocal space, meaning that the magnetic structure is not characterized by a singled-out propagation vector but instead has a manifold of propagation vectors that define the continuous degeneracy of the ground-state. Spins in the SSL state exhibit collective fluctuations, so the SSL-hosting materials were proposed as a promising platform for the experimental realization of emergent excitations such as sub-dimensional fractons, which have restricted mobility and are associated with gauge fields[14,15,17–20]. Materials that were found to exhibit the

[1]Institut für Festkörper- und Materialphysik, Technische Universität Dresden, D-01069 Dresden, Germany. [2]Laboratory for Neutron Scattering and Imaging, PSI Center for Neutron and Muon Sciences, Paul Scherrer Institut, CH-5232 Villigen-PSI, Switzerland. [3]Department for Hydrogen Technology, Institute for Energy Technology, Kjeller NO-2027, Norway. [4]Neutron Scattering Division, Oak Ridge National Laboratory, Oak Ridge, TN 37831, USA. [5]Würzburg-Dresden Cluster of Excellence on Complexity and Topology in Quantum Matter—ct.qmat, TU Dresden, Dresden, Germany. [6]Experimental Physics VI, Center for Electronic Correlations and Magnetism, University of Augsburg, 86159 Augsburg, Germany. [7]Max Planck Institute for Chemical Physics of Solids, D-01187 Dresden, Germany. ✉e-mail: nikita.andriushin@tu-dresden.de; Michael.Baenitz@cpfs.mpg.de; aleksandr.sukhanov@tu-dresden.de

SSL signatures so far are the honeycomb compound FeCl$_3$[21], the pyrochlore ZnCr$_2$Se$_4$[22–24], the breathing-kagome lattice crystal Ca$_{10}$Cr$_7$O$_{28}$[25], and the diamond-lattice material MnSc$_2$S$_4$[26,27]. In addition, the compound LiYbO$_2$ with an elongated diamond lattice was proposed as an SSL based on the powder-averaged data[28]. The triangular lattice is, in turn, the prototype of a geometrically frustrated lattice, which was also predicted to host the SSL state[29,30] yet, no experimental realizations of the SSL on a triangular lattice were reported so far.

In Fig. 1b we show the magnetic phase diagram of the triangular lattice with the $J_1 - J_2 - J_3$ Heisenberg exchange interactions [Fig. 1(a)] with a ferromagnetic (FM) $J_1$ (negative) and antiferromagnetic (AFM) $J_2$ and $J_3$ (both positive). To improve the visual clarity, the phase diagram is decorated by the color-coded magnitude of the propagation vector **q**, which depends on the relative ratio of exchange interactions, $J_2/J_1$ and $J_3/J_1$ (see Section S2 in Supplemental Materials[31]). In addition to the trivial FM order and commensurate $(\frac{1}{3}\frac{1}{3}0)$ AFM states, two spiral ground states can be realized: the phase II, where the propagation vector is aligned in reciprocal space as **q** = $(\xi\,0\,0)$ [Fig. 1(c3)], and the phase III, for which the spiral propagates along an alternative direction **q** = $(\xi\,\xi\,0)$ [Fig. 1(c1)]. In other words, the phases II and III describe the same spin spiral but oriented along each of the two principal crystallographic directions. The most interesting part is the boundary between the two spiral phases defined by $J_2/J_3 = 2$, for which the two orientations of propagation wavevector become degenerate. At the critical line, the spiral propagation vector is no longer bound to the underlying lattice, and thus can arbitrarily rotate (but keeping its magnitude) on the two-dimensional plane [see the cartoon in Fig. 1(c2)]. This leads to the characteristic singularities in its structure factor directly probed by neutron scattering, namely, to the emergence of a continuous ring of scattering intensity in reciprocal space with the radius set by the same ratio of the exchange parameters $J_2/J_3$.

Here we demonstrate that AgCrSe$_2$, a layered compound with the crystal structure similar to the delafossite family[32–37], is a perfect realization of the triangular lattice with the nearest-neighbor FM exchange and the next-nearest-neighbor AFM exchange interactions. The material has a trigonal crystal structure, where the magnetic Cr$^{3+}$

ions ($S$ = 3/2) form triangular layers stacked in the ABC-fashion [Fig. 1(d)]. Unlike the delafossites, the Ag ions in AgCrSe$_2$ occupy only one of two triangular sublattices, leading to a non-centrosymmetric $R3m$ structure. Although the layers are weakly coupled by an AFM interaction along the trigonal $c$ axis, the physics of each triangular layer governed by the phase diagram of Fig. 1(b) remains intact.

The bulk properties of AgCrSe$_2$ are comprehensively discussed in a previous study[35]. While a broad maximum was observed in the magnetic susceptibility measurements, no signatures of any phase transition could be seen in the specific heat. Instead, the extracted magnetic contribution to the total specific heat is a very broad hump with a maximum centered at $T_X$ = 43 K, suggesting a build-up of strong magnetic fluctuations without long-range order[35]. On the other hand, powder neutron diffraction at 1.5 K showed clear magnetic Bragg peaks that are sharp in the $2\theta$ angle, implying existence of magnetic correlations with a well-defined period. The Bragg peaks were indexed by an incommensurate propagation vector **Q** = (0.037 0.037 3/2)[35] corresponding to the spiral spin correlations akin to the phase III of Fig. 1(b).

In this work, we resolved the existing controversies by employing single-crystal elastic neutron scattering, in which the magnetic intensities can by fully mapped out in 3D momentum space. We show that the anomalies at $T_X$ occur at a cross-over from a weakly-correlated paramagnetic state to the SSL. We further corroborate our experimental observations by spin-dynamics simulations that reproduce the SSL state and its neutron scattering intensity distribution in reciprocal space, including its behavior upon temperature variation and applied magnetic field.

## Results
### Observation of the SSL state
We begin presenting our single-crystal neutron diffraction results with a brief overview of the geometry in reciprocal space. Interacting Cr spins in AgCrSe$_2$ reside on the vertices of the triangular lattice in the $ab$ plane. The spins in individual triangular layers align strictly antiparallel due to an additional AFM exchange along $c$, leading to an out-of-plane component to the propagation vector, i.e. **Q** = **q**$_m$ + $(00l)$, where $l$ = 3/2.

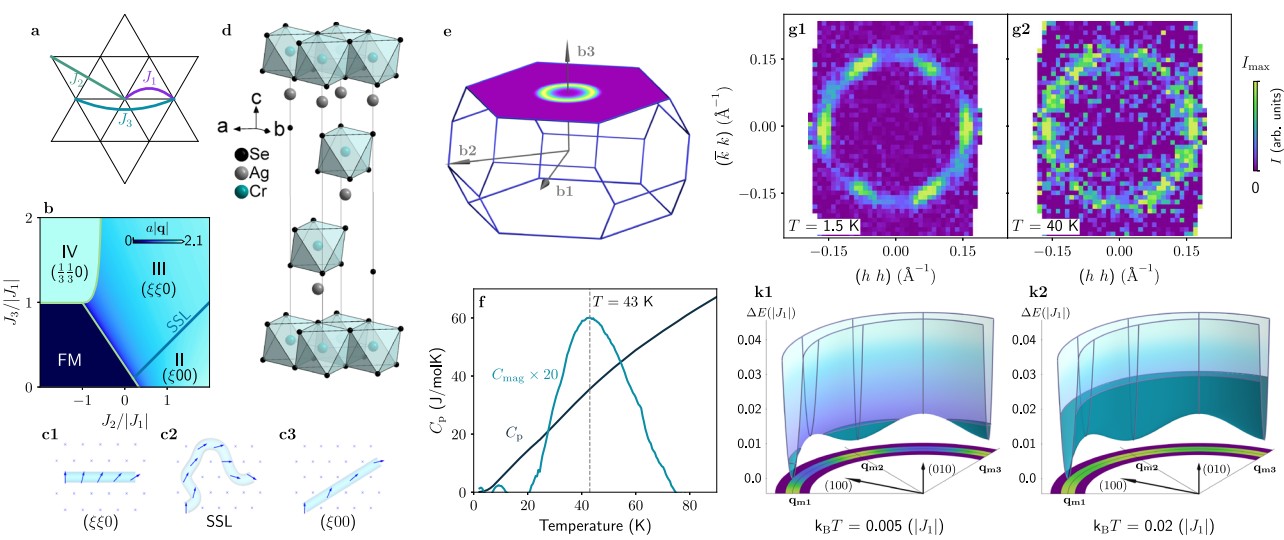

**Fig. 1 | The spiral spin liquid state in AgCrSe$_2$. a** Scheme of intralayer exchange interactions up to third-nearest neighbor. **b** Classical zero-temperature phase diagram in coordinates $J_2/|J_1|$ and $J_3/|J_1|$, the color represents the magnitude of the propagation vector[30]. (c1–c3) Schematic illustration of the cycloidal magnetic order in phase III (c1), phase II (c3) and the SSL state at the boundary between II and III (c2). Crystal structure (**d**) and Brillouin zone (**e**) of AgCrSe$_2$. **f** Temperature dependency of heat capacity of AgCrSe$_2$[35]. (g1,g2) The SANS-I data measured in AgCrSe$_2$ at two temperatures. Note that the measured data include **q**-points with $h > -0.04$, which were symmetrized to visually illustrate the angular intensity distribution. (k1,k2) The classical energy calculated for exchange parameters corresponding to **q**$_m$ = (0.2 0.2 0) r.l.u. and $J_2/J_3$ = 1.7 as a function of the propagation vector of the spin cycloid. The filled part of the surface shows the level of thermal energy $k_B T$. The colormap at the bottom shows the emergence probability of the spiral state with associated propagation vector in a form of $\exp(-\Delta E/k_B T)$.

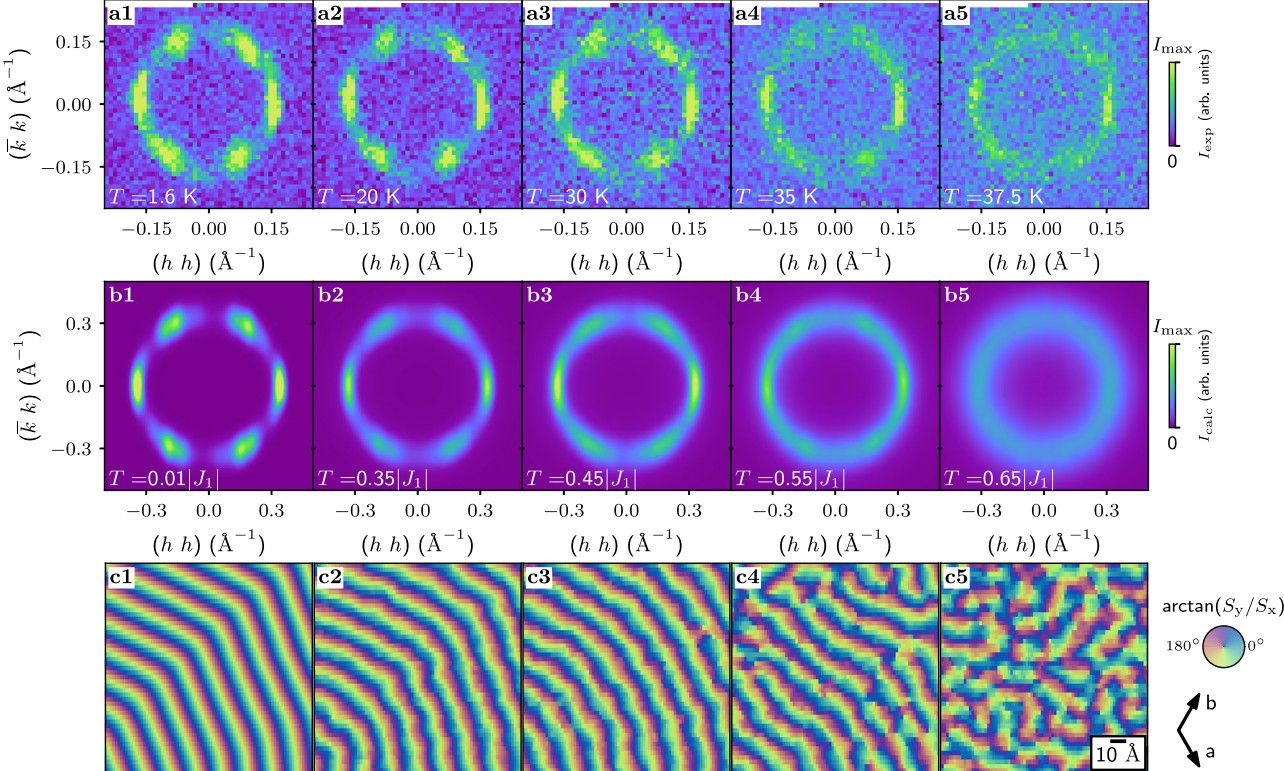

**Fig. 2 | Neutron diffraction in AgCrSe₂. a1–a5** The diffraction maps measured using DMC instrument at different temperatures, as indicated in each panel. **b1–b5** The calculated structure factor. **c1–c5** The real space spin configurations corresponding to the structure factors in (**b1–b5**) panels. The color represents the in-plane angle of the spins.

This shifts the magnetic reflections to the $(H\,K\,3/2)$ reciprocal plane of the first Brillouin zone (BZ), as depicted schematically in Fig. 1e.

The diffraction map collected at $T = 40$ K, just below the maximum of a broad hump in $C_{mag}$ [Fig. 1(f)], displays a continuous ring of magnetic intensity at $|\mathbf{q}_m|$, demonstrating that the ground state of AgCrSe₂ at high temperature is indeed highly degenerate [Fig. 1(g2)]. It is important to note that the uniform ring retains its radial sharpness, indicating that the periodicity of the spiral correlations is well-defined (the radial broadening is limited by the instrumental resolution).

Having confirmed the formation of the SSL state at elevated temperatures, we now turn to the detailed analysis of the scattering intensity upon temperature variation. A series of diffraction patterns in the same reciprocal plane were taken at several temperatures between 1.5 and 37.5 K are shown in Fig. 2(a1–a5). One can notice that the scattering intensity undergoes a smooth transformation upon lowering the sample temperature. Namely, the initially isotropic intensity redistributes into a set of six broad maxima and minima, such that the $\langle 110 \rangle$ directions become favorable for the propagation of the spiral correlations. Because the anisotropy of scattering intensity occurs gradually upon cooling, it is in full agreement with the crossover behavior (as opposed to a phase transition) seen in the specific heat measurements [Fig. 1(f)][35]. As thermal fluctuations diminish, the full SSL degeneracy is lifted, yet the system effectively lacks the long-range order even at $T \ll T_x$ as the spiral propagation is never locked into a singled-out crystallographic direction. We note, that a long-range order, if existed in AgCrSe₂, would manifest itself as resolution-limited Bragg peaks.

The magnetic correlations preserve the spiral periodicity in the full temperature range below $T_x$, as can be seen by the resolution-limited radial width of the scattering intensity. To further quantity the spiral-orientation disorder in AgCrSe₂ at low temperatures, we analyze its azimuthal profiles of intensity in detail. We described the azimuthal

profiles using Lorentzian functions, whose full width at half maximum (FWHM) is proportional to the degree of disorder[38].

Figure 3 (a) summarizes the angular profiles of intensity within the ring, which was plotted by rebinning the diffraction maps into polar coordinates. The maxima spaced by 60° were fitted with a Lorentzian function convoluted with the instrumental resolution (see Section S1 Supplemental Materials[31] for the details on instrumental resolution). The FWHM of the Lorentzians exhibits no change between 2 and 30 K but increases significantly at higher temperatures [Fig. 3(b1)]. As was noted above, the intensity broadening takes place gradually and diverges in proximity to the crossover temperature, where separate Lorentzian profiles are no longer applicable. As can be seen, there is still a significant remnant azimuthal width of ~10° (far surpassing the instrumental resolution) at 1.5 K, associated with a moderate directional disorder of the spirals even at $T \ll T_x$. The reliability of the extracted FWHM can be confirmed by comparing the data collected on two different instruments with distinct resolution functions (see Methods and Supplemental Materials[31]). The extracted FWHMs from the two datasets are plotted along in Fig. 3(b1), confirming that the remnant broadening is an intrinsic property of AgCrSe₂.

It is representative to compare the crossover behavior seen in azimuthal width with the integral intensity of the whole ring. As shown in Fig. 3(b2), the observed intensity displays a gradual decrease upon warming without any sharp anomalies and does not resemble a typical critical $\sim \sqrt{T - T_C}$ behavior that is characteristic of a second-order phase transition. The absence of sharp anomalies was also evident from the magnetic susceptibility measurements in Fig. 3c that show a broad maximum centered at $T_x$[35].

It is worth mentioning that the observed SSL state may explain the recent observation of the anomalous Hall effect in AgCrSe₂[37]. The anomalous part of the transverse resistivity was maximized at the low temperatures but was also observed at temperatures well above $T_X$

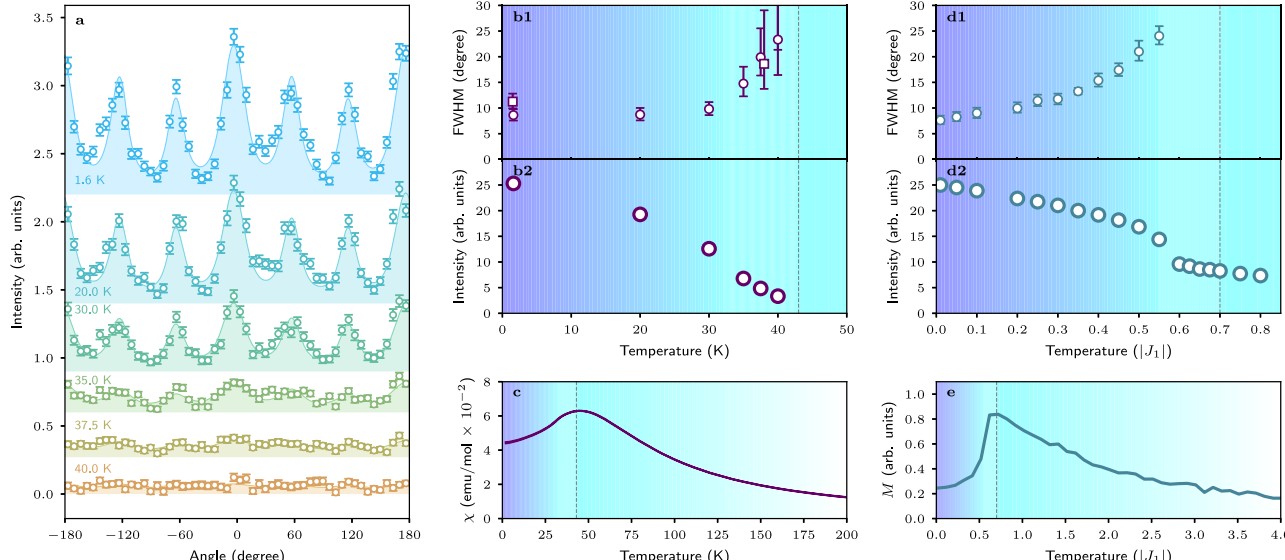

**Fig. 3 | Details of temperature evolution. a** The angular dependency of the intensity (DMC), the color filled peaks are the fitted Lorenzian peaks convoluted with instrumental resolution. The bottom of each color filled peak shows applied offset. **b1, b2** The obtained fit parameters for peaks in panel (**a**): The circles are the Lorentzian FWHM and the intensity extracted from DMC experiment data. The square points show the Lorentzian FWHM in SANS-I experiment, extracted similarly. **c** The susceptibility data measured with in-plane field of 1 T on cooling[35]. **d1, d2** The calculated FWHM and the intensity. **e** The magnetization calculated in field of 0.02$|J_1|$ applied along [110] direction.

without any sharp onset akin to a crossover. This suggest that the inhomogeneous spin texture of the SSL may have a nontrivial impact on the conduction electrons.

## Modeling of the SSL state

To simulate the SSL state in AgCrSe$_2$ we considered the $J_1 − J_2 − J_3$ Heisenberg model with an FM $J_1$ and an AFM $J_2$ and $J_3$. We also included a weak easy-plane anisotropy $K$ that enforces coplanar spin texture with $\langle S_z^2 \rangle = 0$, which was deduced in the previous study[35]. To verify the adequacy of this approach, we first successfully reproduced the magnetic phase diagram of Fig. 2(b1) that was originally obtained by analytic equations[30].

To reproduce the magnetic behavior of AgCrSe$_2$ we chose the following model parameters: $J_2 = 0.33|J_1|$, $J_3 = 0.19|J_1|$ and $K = 0.03|J_1|$. The ground state propagation vector for these parameters equals to (0.2 0.2 0) r.l.u., which is larger than the experimental value. Nevertheless, this parameter set allows us to capture all the general features of the SSL model at reasonable computation time. The larger propagation vector only leads to minor quantitative changes with respect to the experimental values.

The results of our simulations are presented in Fig. 2(b1–b5), where the calculated structure factors are compared to the experimental patterns. For a clear comparison, we convoluted the calculated patterns with the experimental resolution. The simulations were performed at temperatures given in the units of the first exchange $J_1$, as it serves as the overall scaling of the total energy of the system. As one can see, the simulations correctly predict the anisotropic intensity distribution at low temperatures, where the azimuthal broadening shows minor variation from $T = 0.01|J_1|$ to $T = 0.45|J_1|$. The broadening is enhanced at higher $T$, and a fully isotropic ring of intensity is emerging at $T = 0.65|J_1|$ in a good agreement with the experimental pattern at 37.5 K.

We further analyzed the simulated intensities in terms of Lorentzian FWHM and plotted it in Fig. 3(d1) along with the ring integral ring intensity in Fig. 3(d2). We can conclude that the crossover behavior expected for the SSL state is fully realized in the simulations, in excellent agreement with the experimental data. Moreover, the

simulated temperature dependence of magnetization, $M(T)$, closely resembles the experimental data, yet with a somewhat larger drop of the susceptibility at the lowest temperatures. The maximum of the simulated $M(T)$ is found at $T = 0.7|J_1|$. The simulations above $0.7|J_1|$ predict that the SSL state loses its radial correlations and smoothly transforms to what was previously termed a "pancake liquid" (see Section S3, S4 in Supplemental Materials[31])[15,16,39]. Because the intensity of the pancake liquid state is very low, it becomes indistinguishable from instrumental background in our measurements.

It is worth noting that a minor feature identified in the magnetic susceptibility in the previous work[35], namely a maximum in its temperature derivative for the in-plane fields at ~ 32 K corresponds to an accelerated increase of the peak broadening accompanying a small drop of the intensity seen in both the experimental data in Fig. 3(b1, b2) and our simulations in Fig. 3(d1–d2) at $T \approx 0.55K|J_1|$, which further supports the chosen exchange parameters used for the modeling.

Having achieved a good agreement between the simulated and experimental intensities in reciprocal space, we can closely examine the real-space configurations of the spins that correspond to each structure factor. The magnetic texture in real space can be presented through the rotation of the classical spin in the *ab*-plane [Fig. 2(c1–c5)], as the $S_z$ component is mostly zero across the sample plane due to the easy-plane anisotropy. At the lowest temperature [Fig. 2(c1)], the plane consists of two major areas of the cycloidal modulations, where the propagation vector retains its orientation over the distances exceeding 100 Å. The modulations in the two areas are continuously merged such that the spins of the same phase form hexagonal-shaped contours across the sample, as opposed to the distinct domains with domains walls characteristic of the ordered spiral magnets. As the temperature is elevated [Fig. 2(c2, c3)], the hexagonal-shaped spin stripes become more and more concentrically bent, such that the propagation vector is found in one position over the regions less than 100 Å. The cycloid orientation starts fluctuating already on the scale of 10 Å at $T = 0.55|J_1|$ [Fig. 2(c4)]. Finally, the propagation direction is defined only on the scale under 10 Å at $T = 0.65|J_1|$ [Fig. 2(c5)]. This state corresponds to the fully degenerate SSL as opposed to the states with partially lifted degeneracies at low temperatures.

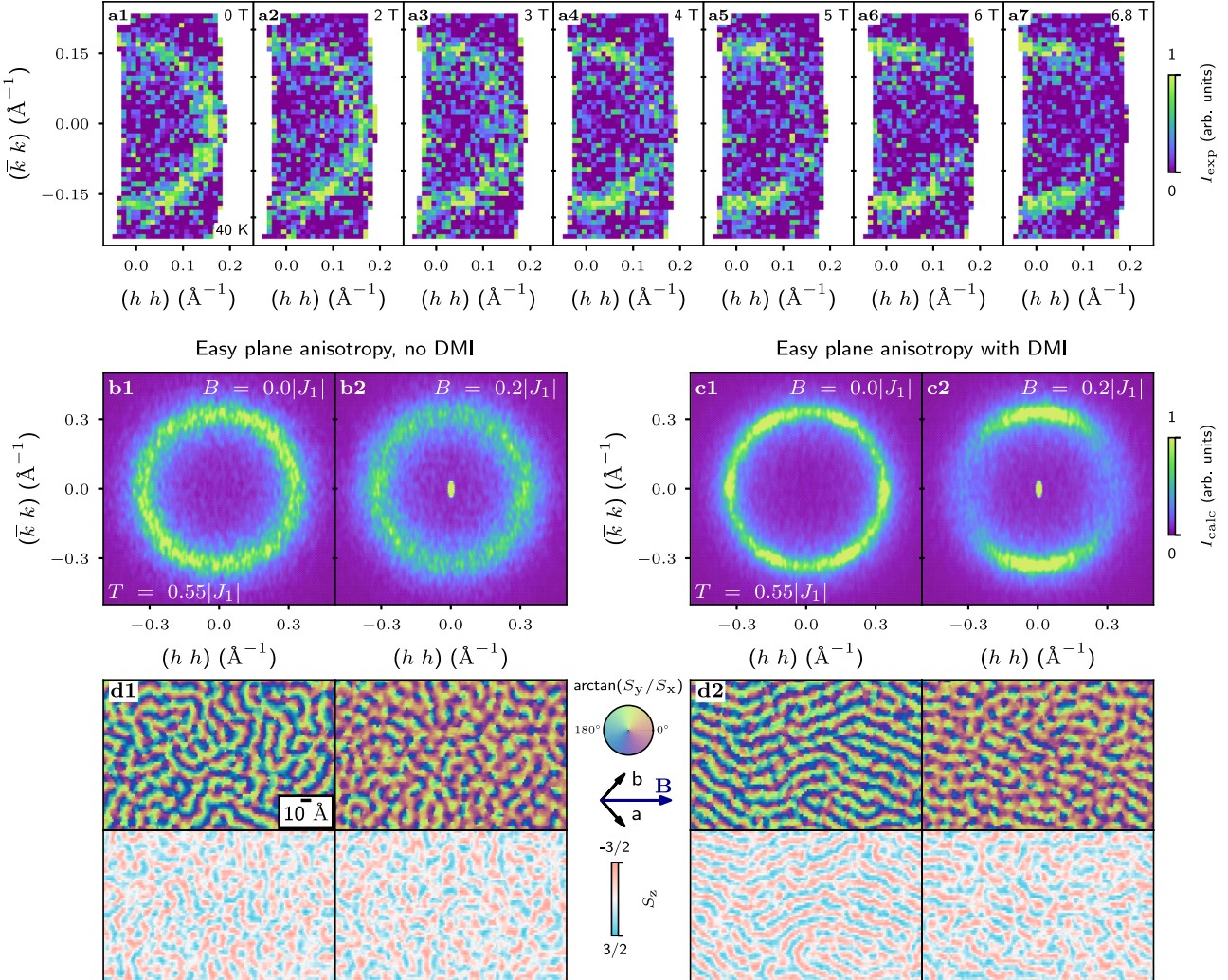

**Fig. 4 | Effect of external field. a1–a7** The diffraction maps measured with SANS-I in different fields along [110] at 40 K. **b1, b2** Comparison of simulated structure factors in absence and presence of external field applied along [11] direction (horizontal to the figure) for the model with easy plane anisotropy only. **c1, c2** Simulations in the same conditions but with introduction of the DMI interaction into the model. **d1, d2** The real space representation of simulated above structure factors. Each column corresponds to the (**b1–c2**) panel above. The top row shows the in-plane orientation of the magnetic moment as $\arctan(S_y/S_x)$, the color circle in-between panels indicates the angle-color correspondence. The bottom row shows the out-of-plane component $S_z$.

## Effect of external field

The SSL state features isotropic angular distribution of the intensity, preserving the rotational symmetry in *ab*-plane. By applying external in-plane magnetic field, one might break the rotation symmetry and potentially tune the SSL properties. However, the spins in AgCrSe$_2$ are confined to the *ab* plane[35], which makes the SSL propagation vectors insensitive to the external field direction (as soon as it is applied in plane). Indeed, the effective Zeeman field averages out to zero due to full spin winding in the *ab* plane. Contrary to this naive expectations, the SSL state in AgCrSe$_2$ is found to respond by a significant redistribution of the intensity. A field applied along (110) at 40 K gradually suppresses the spiral correlations whose propagation vector is aligned with the field $\mathbf{q}_m \| \mathbf{H}$, and favors the ones with the propagation vector being orthogonal to the field $\mathbf{q}_m \perp \mathbf{H}$ [Fig. 4(a1–a7)]. An apparent explanation for such behavior is that the out-of-plane spin component, which was previously precluded by the easy-plane anisotropy, becomes favorable in the applied field. Consequently, the spin correlations are no longer confined in the *ab* plane, and hence can become sensitive to the orientation of the in-plane field. This scenario is further supported by the previous magnetization measurements, where a spin-flop transition was observed at ~5 T[35].

In our simulations, the frustrated $J_1$–$J_2$–$J_3$ model with an easy-plane anisotropy indeed does not show any rotation symmetry breaking in the presence of an external field [Fig. 4(b1, b2)]. The spiral correlations maintain their orientational disorder and the pitch, while the net magnetization is build because of a slight anharmonisity of the spin spiral modulation: the magnetic moments along the spiral propagation hold on for slightly longer at the "preferable angles", as reflected in Fig. 4(d1) by the increased amount of red regions under the applied field. Out-of-plane spin components, as the result of an interplay between the spiral correlations and an external field, can be realized via the Dzyaloshinskii-Moriya interaction (DMI), which is allowed by the $R3m$ symmetry of AgCrSe$_2$. The in-plane DMI vector, which couples the nearest-neighbor spins (the $J_1$ bond), is oriented perpendicular to the bond and was assumed to have the same magnitude as the easy-plane anisotropy, namely 0.03|$J_1$|. In this configuration, the DMI vector favors spirals with the spin rotation plane within the *ac*-plane and the equilibrium orientation of the rotation plane is defined by the balance between the anisotropy and the DMI.

In the absence of a field, a model augmented with the DMI is characterized by the structure factor [Fig. 4(c1)] that is very similar to the one obtained in simulations with only the easy-plane anisotropy

and no DMI. Because the DMI interaction is weak compared to the primary isotropic interactions, it does not affect the SSL behavior, nor breaks the rotational symmetry.

However, clear alteration in the $S_z$ component of spins is evident in the real space images [Fig. 4(d2)]. The DMI causes the $S_z$ component to closely follow the spin spiral propagation. This directly influences the system's response to an external in-plane field: the spiral correlations propagating along the field direction are now disfavored as compared to the rest. The structure factor in applied field [Fig. 4(c2)] then well matches the experimental data [Fig. 4(a)], providing a clear evidence to the presence of DMI in the system.

It is worth noting that the neutron powder diffraction measurements[35] indicated that the cycloidal modulations occur within a plane that is oriented at 89(7)° with respect to the $c$ axis[35], possibly slightly off the $ab$ plane. This sets constraints on the possible strength of the DMI. The DMI is expected to be sufficiently weak not to overcome the easy-plane anisotropy at low temperatures in zero field, while still being strong enough to affect spiral propagation vector in the applied fields, as has been experimentally observed. However, an exact quantitative analysis of the DMI remains beyond the present study.

## Discussion

It is important to note that the zero-temperature phase diagram [in Fig. 1(b)] predicts the SSL ground state exclusively at $J_2/J_3 = 2$, where the maximum level of frustration is realized. However, at finite temperatures, the SSL properties are not limited to the phase boundary of the two incommensurate states and effectively appear due to thermal fluctuations. Since the effect is mainly based on the frustration of exchange interaction, it virtually exists in the entire space of parameters compatible with an incommensurate spiral order. The temperature region where it is realistically observable heavily depends on the level of frustration, namely the proximity to the II/III phase boundary. Quantitative characteristics, such as the SSL temperature range extent and how quickly this regime develops, can change with the $J_2/J_3$ ratio and the magnitude of the propagation vector (for details see the Section S2 of Supplementary Materials[31]). However, the general feature of the SSL state appearance through a crossover is preserved. To further illustrate this, we calculated the Heisenberg classical energy in this model as a function of $\mathbf{q}$, see Fig. 1(k1, k2). Clearly, the energy has a deep minimum at $|\mathbf{q}_m|$, but exhibits only a small difference between $\mathbf{q}\|(110)$ and $\mathbf{q}\|(100)$, which provides an intuitive understanding for our neutron diffraction data and the spin-dynamics calculations at finite temperatures. Essentially, thermal fluctuations first destroy the long-range order by making any orientation of the spin spiral energetically equal, and only at much higher temperatures do they destroy the spiral correlations. A rough estimation of temperature smearing is $\exp(-\Delta E/k_B T)$, which gives a probability to find a spiral state with $E = E_{min} + \Delta E$ in a system thermalized at temperature $T$. The corresponding colormap is shown in the lower part of Fig. 1(k1, k2).

The sizable easy-plane anisotropy in AgCrSe$_2$ effectively restricts its spin dimensionality at low temperatures, which makes the XY spin model applicable for the low-energy physics of this material. Its ground state has the U(1) × U(1) symmetry, where one U(1) describes the spin rotation, whereas another U(1) stems from the momentum rotation on the spiral ring. This contrasts with the O(3) × U(1) symmetry of the Heisenberg model seemingly applicable to the SSL hosts FeCl$_3$[21] and MnSc$_2$S$_4$[26,27]. Because the SSL formation within the XY model has already been quite extensively discussed in the theoretical works[12–20], AgCrSe$_2$ becomes an attractive playground for testing those predictions. For example, Yan and Reuther[15] presented a topological classification of the momentum defects formed in the SSL state. After a close look into the spin configurations stabilized in our simulations, we were able to conclude that the nontopological momentum vortices are indeed realized for the model parameters of AgCrSe$_2$ (see Section 5 of Supplementary Materials[31]). However, the momentum vortices with a non-zero topological charge (as defined in[15]) did not appear in our simulations, which might be related to either their higher energy costs or the fact that the system is simulated on cooling from a high temperature where the XY model is no longer applicable, which may preclude the nucleation of the topological defects. Therefore, application of the emergent higher-rank gauge theory[15] to AgCrSe$_2$ remains an open question worth addressing in future studies.

To summarize, we report the observation of the SSL state by neutron scattering in a material with the perfect triangular lattice, and qualitatively reproduce our observations by the Landau-Lifshitz spin-dynamics simulations. The origin of the SSL, relying on thermal fluctuations and degeneracy, could be unraveled in terms of a specific energy landscape favoring orientational disorder at finite temperatures. The SSL behavior in external magnetic field suggests the presence of non-negligible DMI in AgCrSe$_2$ providing a way of tuning the spiral correlations with an external in-plane field. Our study confirms the validity of the preceding theoretical proposals, and broadens the class of potential SSL-hosting compounds to include the triangular lattice materials, these being one of the most important model systems in the frustrated magnetism.

## Methods

### Sample preparation
A high-quality single crystal of AgCrSe$_2$ with lateral dimensions of a few mm and mass ~ 8 mg was grown by chemical vapor transport using chlorine as transport agent, as described in details in[35]. The sample composition and crystalline quality was characterized by energy-dispersive x-ray spectroscopy, wavelength-dispersive x-ray spectroscopy, x-ray powder diffraction, Laue x-ray diffraction, and differential scanning calorimetry. The bulk magnetic properties were characterized by DC and AC magnetic measurements and specific heat measurements reported in[35].

### Neutron scattering experiments
The neutron diffraction data were collected using the cold neutron diffractometer DMC and the small angle neutron scattering instrument SANS-I, both located in Paul Scherrer Institute (PSI), Switzerland.

In the DMC experiment, we used a PG(002) monochromator to select wavelength $\lambda = 2.45$ Å. The in-plane $\mathbf{Q}$ resolution of DMC is considerable better than the out-of-plane, which effectively elongates the Bragg peaks in the [$\bar{1}$10] direction (perpendicular to the scattering plane). The sample was placed in a standard orange cryostat and rotated around its vertical axis with a small angle step of 0.1°, such that a wide reciprocal-space volume was mapped out by the detector array covering 2$\theta$ of 127° in plane and ± 7° out of plane. The obtained 3D dataset was then sliced in high-symmetry planes for further analysis.

The SANS-I experiment was conducted on the same sample and the same sample orientation with respect to the scattering plane. The magnetic field was applied a using a horizontal 6.8 T cryomagnet (opening angles 45°), with the magnetic field being applied along the incident neutron beam [close to the ($\bar{1}$10) reciprocal direction of the sample]. The sample and magnet were rotated around the vertical axis over 25° range in 0.5° steps to continuously span the momentum range of interest [the magnetic satellites in the ($HK$ 3/2) plane]. The individual diffraction patterns collected at different sample angles by 2D detector of 0.96 × 0.96 m$^2$ were combined into a 3D dataset. The 3D dataset was then sliced in the ($HK$ 3/2) plane for further analysis. Due to the geometry of the SANS-I setup, the resolution ellipsoid is elongated along the (00$L$) reciprocal direction, whereas the resolution in the ($HK$0) plane (the plane of the SSL intensity) is significantly narrower and is approximately twice better than in the DMC measurements. For all the measurements, we subtracted the pattern collected at 50 K as a background, see Section S4 in the Supplementary Information[31].

## Numerical simulations

The classical spin-dynamics simulations were performed with the Landau-Lifshitz dynamics approach as implemented in SU(N)NY program package[40]. A single triangular lattice layer formed with up to $300 \times 300$ dipolar spins and periodic boundary conditions was considered. For a detailed description, see Section S2 in the Supplementary Information[31].

## Data availability

All relevant data are available from the authors upon reasonable request.

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

## Acknowledgements

We thank V. Hasse for technical support in crystal growth. We acknowledge financial support from the Swiss National Science Foundation, from the European Research Council under the grant Hyper Quantum Criticality (HyperQC). N.D.A. and D.S.I. are greatful for support of the German Research Foundation (DFG) through the Collaborative Research Center SFB 1143 (project # 247310070); through the Würzburg-Dresden Cluster of Excellence on Complexity and Topology in Quantum Materials — ct.qmat (EXC 2147, Project No. 390858490). This work is based on experiments performed at the Swiss spallation neutron source SINQ, Paul Scherrer Institute, Villigen, Switzerland. M.C.R. is grateful for support through the Emmy-Noether program of the DFG (project-id 501391385).

## Author contributions

A.S.S, M.B. and S.E.N. conceived the idea and designed the experiments. Crystals were grown by M.S. Neutron scattering was performed by

S.E.N., A.S.S., Ø.S.F., J.S.W., A.P. and the data were analyzed by N.D.A., S.E.N. and A.S.S. Simulations and analysis were performed by N.D.A., A.S.S. and S.E.N. The manuscript was written by A.S.S., N.D.A. and S.E.N with assistance of M.C.R. and D.S.I. and all authors discussed the results and commented the manuscript.

## Funding

## Competing interests

The authors declare no competing interests.
