## [Transparent Peer Review file · Nature Communications]

Observation of the spiral spin liquid in a triangular-lattice material

Corresponding Author: Mr Nikita Andriushin

Version 0:

Reviewer comments:

Reviewer #1

(Remarks to the Author)

In this study Andriushin et al. present their results of an elastic neutron scattering experiment on single crystals of AgCrSe₂ (ACS). Their main finding is the presence of a spiral spin liquid phase in these crystals which persists even for temperatures as high as 40 K. The phase transition is brought into context with T^* of approx. 42 K which has been observed in previous experiments for example as a broad peak in the measured specific heat as cited from Ref 30. If true, these findings, supported by their high quality data, would be of general interest to the community and worth publishing in Nature communication. While I find the paper very interesting and would lean towards publication, I do not yet recommend publishing the article in its current state. In order to do so, I would ask the authors to discuss especially the high temperature phase transition in more detail. This would not just clarify some of the statements made in the paper but also strengthen their argument, however, this discussion is rather ambiguous in the current manuscript (see point 1 for further discussion). The paper could further profit from providing more information of how this relates to some of the existing data at certain places. In addition to this I have a few minor points I would like to be addressed at the end of the review.

1.

An important point is the high T phase transition, which is completely absent in the data. In line 256 of the manuscript the authors state that at a temperature of $T=0.7J_1$ the system should transform into a pancake liquid thus being strongly correlated but with a q-vector which is not yet fixed to the length q_0 . Is it possible to measure for temperatures higher than 40K in order to gain more insight into this transition and as such provide additional evidence for your general conclusion? The authors briefly mention it is not visible in the data due to its low intensity, so I assume this has been attempted. I would strongly suggest still showing these scans. Even if no pancake pattern can be observed it would be desirable to show that at least the ring pattern gets resolved. This is especially important as a ring pattern could also be observed for a strongly correlated paramagnetic state.

2.

In addition to showing this data, you do not discuss the pancake liquid in the supplementary material as stated in the main text but a calculation of what to expect and where this phase transition would be in temperature/Kelvin would be highly desirable. Assuming the transition from SSL to pancake liquid is indeed at 43K I would really like to see an estimate/calculation of the transition temperature where the system undergoes the transition from the pancake liquid to paramagnetic state in Kelvin. The reason for this is, that for the pancake liquid I naively would have expect a signature in the specific heat such as shown here for T^* , it would therefore be interesting to see if there is another hint and thus evidence of this transition in the existing data.

3.

Related to this, I would like a more detailed discussion, which is brought into context with existing literature, on the previous assignment of ACS being an anti-ferromagnet with a Neel temperature of 32 K. The authors mention that the transition between the SSL and spiral ground state is continuous yet there are clear features in previous experiments at 32 K. In previous studies a strong magnetic anisotropy has been observed up to 32K which as well as a small anomaly in the specific heat measurements. Here, there seems to be no significant change in the data, how does this fit together? Are you implying this is where the spiral ground state is fully formed?

Few smaller points:

4.

In addition, a quick literature search showed that ACS has been measured with a number of different techniques. I would be interested about what you would expect to happen to the electronic structure in the SSL state. Is this in line with for example ARPES measurements and the lack of shifting in of the bulk electronic structure shown in a previous study of some of the authors? This also seems to suggest a local moment picture despite the Cr atoms? I would be interested in a brief statement by the authors.

5.

The authors call ACS a delafossite. I would maybe suggest using the word delafossite-like, as the delafossite structure describes a very specific crystal structure which is centro-symmetric. The difference here is the occupation of only one of the two sublattices in the Ag plane which results in the compound not being centro-symmetric. In general, it might be worth at least mentioning the Ag plane here, too. This also is important because you say your phase diagram holds up for these triangular layers, again this is ambiguous because only true for the CrO₂ layers in the crystal.

6.

Furthermore, while the manuscript is interesting, I may suggest some small changes in the writing at places to avoid ambiguity and increase clarity. One example is the propagation vector q . While it is mentioned multiple times it never is stated that in the SSL it is fixed in length to q_0 . It reads a bit like the length can also vary which would of course be what you expect in the pancake liquid instead (especially line 93 might benefit from the addition). As another example; in line 112 and 114 you talk about 'certain anomalies' and 'phase transitions'. Especially for someone who does not know ACS it would be useful to clearly state which anomaly and which phase transition you are talking about this happens more often in the paper and should be resolved so it is better to understand without knowing the compound. In lines 151 and 157 you state that your data 'confirms' the SSL without any further discussion. At this point, there has been no complete proof other than the data but a ring itself can also be evidenced for a strongly correlated paramagnetic system so without any discussion this simply is not enough. As all your discussion and analysis follows after I would thus be careful about the language you are using, trying something a bit less definite such as 'strongly suggests'. These are just few examples/suggestions.

7.

Fig 1 c1 and c2 might benefit from including the 'II' and 'III' labels but this is just a suggestion. For g1 and g2 however, you never introduce SANS as an abbreviation which you might want to consider especially for a reader that does not do Neutron scattering experiments.

Reviewer #2

(Remarks to the Author)

In the submitted manuscript, authors Andriushin et. al reported the presences of a spiral spin liquid in single crystal AgCrSe₂. This was supported by small angle neutron scattering and simulated model. AgCrSe₂ is a layered material known for its thermoelectric and ionic conductivity properties. The recent emergent of SC AgCrSe₂ fabricated by CVT has allowed further studies regarding the spin ordering within the structure. The current manuscript built upon their prior work as published in Phys Rev B 104, 134410 (2021), where they have reported the magnetic phase diagram of the material along side with deriving the magnetic lattice through neutron diffraction. However, they observed an anomalous feature at 43K and assume it to imply it as the presences of a strong frustration. In this work, they present a detailed study into this feature and resolved it with detail SANS measurements. The SANS and simulated data is convincing in showing the formation of the spiral spin liquid and the response to in-plane magnetisation further shows the dynamic response of the SSL.

It is in my opinion that the manuscript submitted offer significant insight into a unique feature and that it should be accepted for publication by the journal.

Reviewer #3

(Remarks to the Author)

Review of NCOMMS-24-64516

Observation of the spiral spin liquid in a triangular-lattice material

Spiral spin liquids are magnetic phases that do not order down to very low temperatures due to a degeneracy of round states in reciprocal space in the form of a ring (in 2D) or surfaces (in 3D). As such, they are highly interesting systems with a non-trivial emergent effective theory.

In this paper, the authors reported that AgCrSe₂ (now referred to as ACS) is an almost perfect realization of spiral spin liquid on a triangular lattice. My overall conclusion is that this work has the potential to be published in Nature Communications, but its current status lacks some novel physics compared to current literature. In what follows, I will comment on three parts: (One) highlights of this work; (Two) why I think there is still a lack of novelty and what can be done to improve it; and (Three) other regular points of revision to be made. I hope my report will help the authors to enhance the science of their paper and reach the standard of Nat. Comm.

One: highlights of this work

1. The material realizations of 2D SSLs are still rare. As the authors mentioned, the FeCl₃ is an example of Heisenberg spins (but without in-plane anisotropy, as far as experiments show). Therefore, it is exciting to have other experimental realizations.

By the way, the authors have missed citing another series of works on Ca₁₀Cr₇O₂₈, which has been shown to be a SSL too. These works should be mentioned. See:

Physical realization of a quantum spin liquid based on a complex frustration mechanism. Nat. Phys. 12, 942–949 (2016)

Spiral Spin Liquid Noise <https://arxiv.org/abs/2405.02075>

2. The experiment and associated standard theoretical/numerical study are done at high quality and thorough. That being said, I have one important point for the authors to address, see part three point 6.

Two: why I think there is still a lack of novelty and what can be done to improve it

This work is the first case of 2D SLL on the triangular lattice, but not the first case of 2D SLL, as I discussed earlier. The observation of the ring and how it evolves with temperature is very similar to that of FeCl₃ published on PRL, and so is the extent of theoretical analysis.

The real space spin pictures are new here --- similar figures from numerics have been shown in arXiv:2405.02075, but that work, still not officially published, should not be used to deny the novelty here. These pictures, especially Fig. 2, have been proposed in theory works before [Low-energy structure of spiral spin liquids. Phys. Rev. Res. 4, 023175 (2022)], however.

So, while discovering a new material is very important, my dissatisfaction is that there has been no further progress in understanding the physics of SSLs compared to what has been done in FeCl₃ and Ca₁₀Cr₇O₂₈.

There are ways to improve this part. There are now a handful of related theories on SSL on the market. Can the authors suggest which one of these theories is (partially) confirmed or denied by the experimental evidence?

For example, (1) the spinon surface theory: Signatures for spinons in the quantum spin liquid candidate Ca₁₀Cr₇O₂₈. Phys. Rev. B 100, 174428 (2019).

(2) Momentum vortex theory: Low-energy structure of spiral spin liquids. Phys. Rev. Res. 4, 023175 (2022).

(3) Ripple state (ordered state at T below the liquid state regime): Ripple state in the frustrated honeycomb-lattice antiferromagnet, Physical Review Letters 123 (5), 057202.

To give an example, I note that a particularly unique feature in ACS is the anisotropic term that locks the spin in-plane at low temperatures. To my knowledge, this particular case has been studied in theories (2) and (3). In (2), a connection of SSL to momentum vortices, emergent higher-rank gauge theory, and hidden four-fold pinch point has been proposed. Indeed, Figs. 2 c1-c3 resemble zoomed-in views of the spin configurations in (2) and (3). Since that theory is classical, it should be testable (e.g., see if the hidden 4 fold pinch point exists) based on the experimental data combined with the numerics of ACS.

I also want to point out that the ground state space is very different for the XY and Heisenberg spin models. The former is $U(1) \times U(1)$, one $U(1)$ for the spin rotation, and one $U(1)$ for the spiral ring; the latter is $O(3) \times U(1)$. This difference will lead to different topological structures of the topological defects. The authors should emphasize that ACS realize the XY spiral spin liquid at low T.

Three: other regular points of revision

1. Cite relevant earlier works when discussing the phase diagram of Fig. 1b. Such as "Generic Spiral Spin Liquids", Front. Phys. 16, 53303 (2021).

2. Fig. 1 (b) caption: $J_2/J_1, J_3/J_1$ should be $J_2/|J_1|, J_3/|J_1|$.

3. Fig. 1(f): there are two curves. Both curves should be explained in the caption and main text.

4. Line 93: typo: can arbitrary rotate -> can arbitrarily rotate

5. Fig. 2 c1-c3: I wonder if the momentum vortex is observed in the numerics. What is shown here seems to be about a quarter of such a topological defect. Periodical boundary conditions may be better to see such physics.

6. [Important] Why is the six-fold rotational symmetry broken in Figs. 2 b1-b4, computed from the numerics? The intensity along the x-axis is stronger, and the other four dots are weaker. However, the model is six-fold rotationally symmetric. Did the authors choose some symmetry-breaking boundary conditions? This should be explained. Or, if the numerics is not done perfectly, the results should be revised/corrected.

Related to this, the experimental data in Figs. 2a and 3a show the same way symmetry breaking, too, whose physical origin should be explained/speculated.

At the present stage, I am not convinced that the symmetry-breakings in experiments and numerics have the same origin. This is a rather serious question that requires an answer.

Version 1:

Reviewer comments:

Reviewer #1

(Remarks to the Author)

I have carefully reviewed the authors response and the changes made to the manuscript. Most importantly, I find that the revisions regarding the high-temperature data and its connection with existing results have been adequately addressed and incorporated into the document, including new figures and text sections.

While the c1 and c2 labels in Fig. 1 do not appear to have been adjusted as claimed by the authors in their response, my earlier comment regarding these labels was merely a suggestion so not a decision making point. Furthermore, I believe that the new discussion, which follows the recommendations of referee three, adds constructively to the revised manuscript.

In conclusion, I believe that the revised manuscript meets the necessary standards for publication in Nature Communications.

Reviewer #3

(Remarks to the Author)

The updated manuscript has addressed all the key questions I asked. Overall, this experimental work is of high quality, discovers a novel spin system, and has a decent amount of discussion on the theory. Therefore, I recommend its publication on Nat. Comm.

Reviewer 1 (Remarks to the Author):

In this study Andriushin et al. present their results of an elastic neutron scattering experiment on single crystals of AgCrSe₂ (ACS). Their main finding is the presence of a spiral spin liquid phase in these crystals which persists even for temperatures as high as 40 K. The phase transition is brought into context with T^* of approx. 42 K which has been observed in previous experiments for example as a broad peak in the measured specific heat as cited from Ref 30. If true, these findings, supported by their high quality data, would be of general interest to the community and worth publishing in Nature communication. While I find the paper very interesting and would lean towards publication, I do not yet recommend publishing the article in its current state. In order to do so, I would ask the authors to discuss especially the high temperature phase transition in more detail. This would not just clarify some of the statements made in the paper but also strengthen their argument, however, this discussion is rather ambiguous in the current manuscript (see point 1 for further discussion). The paper could further profit from providing more information of how this relates to some of the existing data at certain places. In addition to this I have a few minor points I would like to be addressed at the end of the review.

We are very pleased to see that the Referee finds our manuscript of general interest and worth publishing. We would like to thank the Referee for their time and efforts they put to provide highly relevant and detailed comments. We believe that by addressing all the questions, we significantly enhance the presentation of our results, and hope that the Referee will consider the improved manuscript fully satisfactory.

1. An important point is the high T phase transition, which is completely absent in the data. In line 256 of the manuscript the authors state that at a temperature of $T=0.7J_1$ the system should transform into a pancake liquid thus being strongly correlated but with a q-vector which is not yet fixed to the length q_0 . Is it possible to measure for temperatures higher than 40K in order to gain more insight into this transition and as such provide additional evidence for your general conclusion? The authors briefly mention it is not visible in the data due to its low intensity, so I assume this has been attempted. I would strongly suggest still showing these scans. Even if no pancake pattern can be observed it would be desirable to show that at least the ring pattern gets resolved. This is especially important as a ring pattern could also be observed for a strongly correlated paramagnetic state.

We thank the Referee for pointing out on a somewhat missing discussion of the pancake state above the cross-over temperature T_X (43 K experimental and $0.7J_1$ theoretical). Firstly, we would like to apologize that we completely forgot to mention anywhere in the text that we used a high temperature measurement for dealing with a relatively high unavoidable background in our experiments. We correct this, thanks to the referee. We explicitly mention it now in "METHODS", we quote it here for convenience:

For all the measurements, we subtracted the pattern collected at 50 K as a background, see Sec. S3 in the Supplementary Information.

Accordingly, we added a new supplementary section "High temperature data and background subtraction in the SANS-I experiment" (not quoted here), where we present an example of raw data before the background subtraction, the background-subtracted data, and the raw data collected at 50 K. We clarify that weak scattering from a weakly-correlated pancake state is expected on the level of the background at $T > T_X$ and therefore cannot be reliably identified in our measurements. For the same reason, we can indeed safely subtract 50 K from all the data below T_X as it is dominated by the instrumental background. We also note, that the background scattering exhibits a weak T dependence on its own due to the phonon scattering from the sample environment (and the sample as well). Therefore, measurements at much higher temperatures cannot be used for the background subtraction.

2. In addition to showing this data, you do not discuss the pancake liquid in the supplementary material as stated in the main text but a calculation of what to expect and where this phase transition would be in temperature/Kelvin would be highly desirable. Assuming the transition from SSL to pancake liquid is indeed at 43K I would really like to see an estimate/calculation of the transition temperature where the system undergoes the transition from the pancake liquid to paramagnetic state in Kelvin. The reason for this is, that for the pancake liquid I naively would have expect a signature in the specific heat such as shown here for T^* , it would therefore be interesting to see if there is another hint and thus evidence of this transition in the existing data.

We thank the referee for further elaboration of their comment on the pancake state. We would like to remind that the original definition of the pancake state developed in Ref. [T. Shimokawa and H. Kawamura, Phys. Rev. Lett. **123**, 057202 (2019)] refers to an intermediate state between the SSL and a paramagnet that are not connected by any phase transition. Hence, the pancake state describes continuous transformations and no signatures in the specific heat between the strongly correlated SSL and a fully disordered paramagnetic state can be observed [see Fig. 1e in Ref. T. Shimokawa and H. Kawamura (2019)]. Generally, the pancake is characterized by any finite correlations with a maximum at either a finite q or $q = 0$. In this light, we indeed already mentioned it in the previous version of the supplementary information. Namely, in the last paragraph of S2 (CALCULATION DETAILS), which now contains an extended discussion:

The structure factor of the paramagnetic state with correlations at finite propagation vector is demonstrated in Figs. ?? where it is shown for the parameters $J_2/J_3 = 0$ (no frustration) and $J_2/J_3 = 2$ (maximal frustration). As can be seen, this high-temperature state is characterized by a broad ring of intensity regardless of the frustration ratio. Unlike the SSL state, which has a well-defined static spin-spiral length, the correlated paramagnet in Figs. ?? represent spin fluctuations over a wide range of wavelengths.

We would also like to note that the pancake state (Fig. S7) is predicted in simulations even in the complete absence of frustration when $J_2/J_3 = 0$, for which no SSL state exists and the system undergoes a second order phase transition into the long-range spiral order. That shows that the pancake state is rather a subclass of a trivial paramagnet with finite correlations, hence not inherently connected to the physics of the SSL.

To provide an estimate at what temperature the pancake state becomes indistinguishable from a paramagnet in AgCrSe₂, we conducted additional simulations at higher temperatures. We added a new paragraph in the end of S2 saying:

At even higher temperatures, the broad ring gradually transforms into uncorrelated state [Fig. S8(a,b)]. An intermediate regime with finite residual correlations are referred in literature as “the pancake state”. In our simulations, these correlations are generally weak at $T \gtrsim 3|J_1|$ and diminish to full extent at temperatures above $T \gtrsim 20|J_1|$ [Fig. S8(c)].

3. Related to this, I would like a more detailed discussion, which is brought into context with existing literature, on the previous assignment of ACS being an anti-ferromagnet with a Neel temperature of 32 K. The authors mention that the transition between the SSL and spiral ground state is continuous yet there are clear features in previous experiments at 32 K. In previous studies a strong magnetic anisotropy has been observed up to 32K which as well as a small anomaly in the specific heat measurements. Here, there seems to be no significant change in the data, how does this fit together? Are you implying this is where the spiral ground state is fully formed?

We thank the Referee for the comment on the preceding characterization paper on AgCrSe₂ [M. Baenitz et al., Phys. Rev. B **104**, 134410 (2021)]. The seeming contradiction is stemming from the fact that the authors of Ref. M. Baenitz et al. (2021) (some of them are also co-authors in the present manuscript) made their conclusions from the data available at that time on hand. Namely, the powder neutron diffraction measurements [Section 3F of Ref. M. Baenitz et al. (2021)] are unable to distinguish the SSL state from a trivial long-range spiral order due to powder averaging of all the crystal orientations. As the powder neutron diffraction revealed a magnetic Bragg peak, it was erroneously (but very reasonably considering the data at hands) attributed to a long-range order. Therefore, weak features in the magnetic susceptibility at 32 K were tentatively attributed to a phase transition into a long-range state (following the powder neutron results), which apparently contradicted to absence of any anomalies at 32 K in the specific heat.

Only by single-crystal neutron diffraction were we able to discover the true nature of the magnetic state in AgCrSe₂, consistently explaining the specific heat data, namely a very broad hump centered at 43 K and no features at 32 K.

We note, that the difference observed in the magnetic susceptibilities when the field is applied $\parallel c$ and $\perp c$ is a trivial effect of the single-ion magnetic anisotropy playing a bigger role at the lower temperatures and having a diminishing contribution towards the crossover T_X .

Nevertheless, it can be seen in our data in Figs. 3(b1-b2) and (d1-d2) that at $T \approx 32$ K (data) and $T = 0.55J_1$ (simulations) the peak broadening exhibits a somewhat accelerating increase, whereas the overall intensity shows a down-stepping drop. This explains the maximum of the derivative $d\chi/dT$ for $H \perp c$ in Ref. M. Baenitz et al. (2021), which is also reproduced in our simulations in Fig.3 (e).

To clarify this in the manuscript, we added the following text towards the end of **Modeling of the SSL state** (second paragraph from the end):

It is worth noting that a minor feature identified in the magnetic susceptibility in the previous work [M. Baenitz et al. (2021)], namely a maximum in its temperature derivative for the in-plane fields at ~ 32 K corresponds to an accelerated increase of the peak broadening accompanying a small drop of the intensity seen in both the experimental data in Figs. 3(b1–b2) and our simulations in Figs. 3(d1–d2) at $T \approx 0.55K|J_1|$, which further supports the chosen exchange parameters used for the modeling.

Few smaller points: 4. In addition, a quick literature search showed that ACS has been measured with a number of different techniques. I would be interested about what you would expect to happen to the electronic structure in the SSL state. Is this in line with for example ARPES measurements and the lack of shifting in of the bulk electronic structure shown in a previous study of some of the authors? This also seems to suggest a local moment picture despite the Cr atoms? I would be interested in a brief statement by the authors.

We thank the Referee for the interesting question. We would like to mention that the electronic DOS are already comprehensively discussed in Ref. M. Baenitz et al. (2021) from both the experimental and theoretical points of view, which unambiguously point to the localized nature of Cr ions. This further supports our localized moments simulations. Besides, the ARPES study of Ref. [G. R. Siemann et al., npj QM 8, 61 (2023)] only discusses the surface electronic states, which are not relevant to our bulk study. We suspect that the terminated surface may alter the exchange interactions and preclude the detailed frustration balance required for the SSL, thus any relations between the bulk SSL and the surface ARPES would be too speculative.

Instead, we added a comment on the recent electronic transport properties of AgCrSe_2 published in Ref. [S. J. Kim et al., Adv. Sci. 11, 2307306 (2024)]. We added the following text at the end of **Observation of the SSL state**:

It is worth mentioning that the observed SSL state may explain the recent observation of the anomalous Hall effect in AgCrSe_2 [M. Baenitz et al. (2021)] The anomalous part of the transverse resistivity was maximized at the low temperatures but was also observed at temperatures well above T_x without any sharp onset akin to a crossover. This suggest that the inhomogeneous spin texture of the SSL may have a nontrivial impact on the conduction electrons.

5. The authors call ACS a delafossite. I would maybe suggest using the word delafossite-like, as the delafossite structure describes a very specific crystal structure which is centro-symmetric. The difference here is the occupation of only one of the two sublattices in the Ag plane which results in the compound not being centro-symmetric. In general, it might be worth at least mentioning the Ag plane here, too. This also is important because you say your phase diagram holds up for these triangular layers, again this is ambiguous because only true for the CrO_2 layers in the crystal.

We thank the referee for pointing this out. We made the corresponding changes.

6. Furthermore, while the manuscript is interesting, I may suggest some small changes in the writing at places to avoid ambiguity and increase clarity. One example is the propagation vector q . While it is mentioned multiple times it never is stated that in the SSL it is fixed in length to q_0 . It reads a bit like the length can also vary which would of course be what you expect in the pancake liquid instead (especially line 93 might benefit from the addition). As another example; in line 112 and 114 you talk about ‘certain anomalies’ and ‘phase transitions’. Especially for someone who does not know ACS it would be useful to clearly state which anomaly and which phase transition you are talking about this happens more often in the paper and should be resolved so it is better to understand without knowing the compound. In lines 151 and 157 you state that your data ‘confirms’ the SSL without any further discussion. At this point, there has been no complete proof other than the data but a ring itself can also be evidenced for a strongly correlated paramagnetic system so without any discussion this simply is not enough. As all your discussion and analysis follows after I would thus be careful about the language you are using, trying something a bit less definite such as "strongly suggests". These are just few examples/suggestions.

We took the notes on the ambiguities the Referee mentioned. We improved it everywhere accordingly, except for the q_0 suggestion. In the introductory part we clearly stated $|q|$ is a variable that is balanced out by the exchange

parameters [phase diagram of Fig. 1(b)]. Therefore, it is fixed once the exchange interactions are fixed for a material. When we refer to the fixed propagation vector of the SSL in AgCrSe_2 , we used the label $|q_m|$, which has the given magnitude. We believe that our notations are unambiguous and introduction of another label, such as q_0 will be redundant.

We are somewhat puzzled by the referee's comment: "At this point, there has been no complete proof other than the data, but a ring itself can also be evidenced for a strongly correlated paramagnetic system. . .". The sharp ring is widely recognized as a definitive signature of the SSL state and is generally regarded as the 'smoking gun' [Nature Physics 13, 157–161 (2017); Phys. Rev. Lett. 128, 227201 (2022)]. This also aligns with the opinion of Reviewer 3: "Spiral spin liquids are magnetic phases that do not order down to very low temperatures due to a degeneracy of round states in reciprocal space in the form of a ring (in 2D) or surfaces (in 3D)." The spin-spin correlation function directly reveals the degeneracy of spiral states, providing robust evidence for the SSL state. While we acknowledge that additional techniques for analyzing the SSL state in AgCrSe_2 could further enrich the study, we firmly believe that the data we present are sufficient to substantiate our claims.

We hope the Referee will find the updated version unambiguous and well explained.

7. Fig 1 c1 and c2 might benefit from including the 'II' and 'III' labels but this is just a suggestion. For g1 and g2 however, you never introduce SANS as an abbreviation which you might want to consider especially for a reader that does not do Neutron scattering experiments.

We thank the Referee for pointing this out. We edited it as suggested. We are sorry for a confusion. SANS refers to the name of the instrument, which should actually read SANS-I (the data collected at this instrument is then referred as the SANS-I data), we have corrected for this. In METHODS we explicitly mention that the name of the instrument originates from the name of technique "small-angle neutron scattering".

Reviewer 2 (Remarks to the Author):

In the submitted manuscript, authors Andriushin et. al reported the presences of a spiral spin liquid in single crystal AgCrSe_2 . This was supported by small angle neutron scattering and simulated model. AgCrSe_2 is a layered material known for its thermoelectric and ionic conductivity properties. The recent emergent of SC AgCrSe_2 fabricated by CVT has allowed further studies regarding the spin ordering within the structure. The current manuscript built upon their prior work as published in Phys Rev B 104, 134410 (2021), where they have reported the magnetic phase diagram of the material along side with deriving the magnetic lattice through neutron diffraction. However, they observed an anomalous feature at 43K and assume it to imply it as the presences of a strong frustration. In this work, they present a detailed study into this feature and resolved it with detail SANS measurements. The SANS and simulated data is convincing in showing the formation of the spirial spin liquid and the response to in-plane magnetisation further shows the dynamic response of the SSL. It is in my opinion that the manuscript submitted offer significant insight into a unique feature and that it should be accepted for publication by the journal.

We cordially thank the Referee for supporting the publication of our results.

Reviewer 3 (Remarks to the Author):

Review of NCOMMS-24-64516 Observation of the spiral spin liquid in a triangular-lattice material

Spiral spin liquids are magnetic phases that do not order down to very low temperatures due to a degeneracy of round states in reciprocal space in the form of a ring (in 2D) or surfaces (in 3D). As such, they are highly interesting systems with a non-trivial emergent effective theory. In this paper, the authors reported that AgCrSe_2 (now referred to as ACS) is an almost perfect realization of spiral spin liquid on a triangular lattice. My overall conclusion is that this work has the potential to be published in Nature Communications, but its current status lacks some novel physics compared to current literature. In what follows, I will comment on three parts: (One) highlights of this work; (Two) why I think there is still a lack of novelty and what can be done to improve it; and (Three) other regular points of revision to be made. I hope my report will help the authors to enhance the science of their paper and reach the standard of Nat. Comm.

We thank the Referee for a very detailed and scrutinized evaluation of our work. We were inspired to see that the referee not only provided a healthy and balanced criticism, but also presented a well-structured and elaborated opinion on how our manuscript can be made even stronger and more appealing.

One: highlights of this work 1. The material realizations of 2D SSLs are still rare. As the authors mentioned, the FeCl_3 is an example of Heisenberg spins (but without in-plane anisotropy, as far as experiments show). Therefore, it is exciting to have other experimental realizations.

By the way, the authors have missed citing another series of works on $\text{Ca}_{10}\text{Cr}_7\text{O}_{28}$, which has been shown to be a SSL too. These works should be mentioned. See: Physical realization of a quantum spin liquid based on a complex frustration mechanism. *Nat. Phys.* 12, 942–949 (2016) Spiral Spin Liquid Noise <https://arxiv.org/abs/2405.02075>

2. The experiment and associated standard theoretical/numerical study are done at high quality and thorough. That being said, I have one important point for the authors to address, see part three point 6.

We thank the Referee for the positive assessment of our experimental and theoretical work. We find the suggested references highly relevant and we now included them into the introduction.

Two: why I think there is still a lack of novelty and what can be done to improve it This work is the first case of 2D SLL on the triangular lattice, but not the first case of 2D SLL, as I discussed earlier. The observation of the ring and how it evolves with temperature is very similar to that of FeCl_3 published on PRL, and so is the extent of theoretical analysis. The real space spin pictures are new here — similar figures from numerics have been shown in arXiv:2405.02075, but that work, still not officially published, should not be used to deny the novelty here. These pictures, especially Fig. 2, have been proposed in theory works before [Low-energy structure of spiral spin liquids. *Phys. Rev. Res.* 4, 023175 (2022)], however. So, while discovering a new material is very important, my dissatisfaction is that there has been no further progress in understanding the physics of SSLs compared to what has been done in FeCl_3 and $\text{Ca}_{10}\text{Cr}_7\text{O}_{28}$.

There are ways to improve this part. There are now a handful of related theories on SSL on the market. Can the authors suggest which one of these theories is (partially) confirmed or denied by the experimental evidence? For example, (1) the spinon surface theory: Signatures for spinons in the quantum spin liquid candidate $\text{Ca}_{10}\text{Cr}_7\text{O}_{28}$. *Phys. Rev. B* 100, 174428 (2019). (2) Momentum vortex theory: Low-energy structure of spiral spin liquids. *Phys. Rev. Res.* 4, 023175 (2022). (3) Ripple state (ordered state at T below the liquid state regime): Ripple state in the frustrated honeycomb-lattice antiferromagnet, *Physical Review Letters* 123 (5), 057202.

To give an example, I note that a particularly unique feature in ACS is the anisotropic term that locks the spin in-plane at low temperatures. To my knowledge, this particular case has been studied in theories (2) and (3). In (2), a connection of SSL to momentum vortices, emergent higher-rank gauge theory, and hidden four-fold pinch point has been proposed. Indeed, Figs. 2 c1-c3 resemble zoomed-in views of the spin configurations in (2) and (3). Since that theory is classical, it should be testable (e.g., see if the hidden 4 fold pinch point exists) based on the experimental data combined with the numerics of ACS.

We thank the Referee for giving us the hints on a deeper discussion of our results. This led us to discovery of two types of the momentum defects, one of which is found in lines with the XY model predictions given by Yan and Reuther [*Phys. Rev. Res.* 4, 023175 (2022)], whereas the second type is its surprising partner related by the $U(1)$ symmetry.

We demonstrate our new findings on the momentum vortices in details in a new supplementary section S5 “Low-temperature defects of the SSL” and properly discuss it within the DISCUSSION section in the main text. We cite it here for convenience:

The sizable easy-plane anisotropy in AgCrSe_2 effectively restricts its spin dimensionality at low temperatures, which makes the XY spin model applicable for the low-energy physics of this material. Its ground state has the $U(1) \times U(1)$ symmetry, where one $U(1)$ describes the spin rotation, whereas another $U(1)$ stems from the momentum rotation on the spiral ring. This contrasts with the $O(3) \times U(1)$ symmetry of the Heisenberg model seemingly applicable to the SSL hosts FeCl_3 and MnSc_2S_4 . Because the SSL formation within the XY model has already been quite extensively discussed in the theoretical works, AgCrSe_2 becomes an attractive playground for testing those predictions. For example, Yan and Reuther [*Phys. Rev. Res.* 4, 023175 (2022)] presented a topological classification of the momentum defects formed in the SSL state. After a close look into the spin configurations

stabilized in our simulations, we were able to conclude that the nontopological momentum vortices are indeed realized for the model parameters of AgCrSe_2 . However, the momentum vortices with a non-zero topological charge (as defined in [Phys. Rev. Res. 4, 023175 (2022)]) did not appear in our simulations, which might be related to either their higher energy costs or the fact that the system is simulated on cooling from a high temperature where the XY model is no longer applicable, which may preclude the nucleation of the topological defects. Therefore, application of the emergent higher-rank gauge theory [Phys. Rev. Res. 4, 023175 (2022)] to AgCrSe_2 remains an open question worth addressing in future studies.

As to the other theoretical proposals such as the ripple state of Physical Review Letters 123 (5), 057202, we would like to point out that it was considered for the Heisenberg model, and only for the applied magnetic field. As it is not directly relevant to our system, we would like to avoid discussing it, as it would unnecessarily lengthen the manuscript text and might distract the reader from the primary message of our manuscript.

The spinon Fermi surface theory of Phys. Rev. B 100, 174428 (2019) was developed for a quantum spin $S = 1/2$ of Cr^{5+} , where quantum effects are expected to be strong. In contrast, in AgCrSe_2 Cr ions have valence 3+ and therefore have larger spin of $S = 3/2$. For such spin the quantum effects are expected to be much less relevant, and thus we believe that the classical Landau-Lifshitz-Gilbert equation approach is sufficient to capture the essential physics of AgCrSe_2 . We hope the Referee will not consider this as a drawback of the updated manuscript.

I also want to point out that the ground state space is very different for the XY and Heisenberg spin models. The former is $U(1) \times U(1)$, one $U(1)$ for the spin rotation, and one $U(1)$ for the spiral ring; the latter is $O(3) \times U(1)$. This difference will lead to different topological structures of the topological defects. The authors should emphasize that ACS realize the XY spiral spin liquid at low T.

We thank the Referee for pointing this important aspect out. It is now stressed out in our discussion on page 7.

Three: other regular points of revision 1. Cite relevant earlier works when discussing the phase diagram of Fig. 1b. Such as "Generic Spiral Spin Liquids", Front. Phys. 16, 53303 (2021).

We included the citations as suggested.

2. Fig. 1 (b) caption: $J_2/J_1, J_3/J_1$ should be $J_2/|J_1|, J_3/|J_1|$.

Fixed.

3. Fig. 1(f): there are two curves. Both curves should be explained in the caption and main text.

We added the missing description in the caption and the main text.

4. Line 93: typo: can arbitrary rotate -> can arbitrarily rotate

Corrected.

5. Fig. 2 c1-c3: I wonder if the momentum vortex is observed in the numerics. What is shown here seems to be about a quarter of such a topological defect. Periodical boundary conditions may be better to see such physics.

Actually, the periodic boundary conditions were implemented in our simulations. Figs. 2 (c1–c5) are zoom-in's within a larger simulated system. The real space configurations are now analyzed and discussed in details as we elaborated above.

6. [Important] Why is the six-fold rotational symmetry broken in Figs. 2 b1-b4, computed from the numerics? The intensity along the x-axis is stronger, and the other four dots are weaker. However, the model is six-fold rotationally symmetric. Did the authors choose some symmetry-breaking boundary conditions? This should be explained. Or, if the numerics is not done perfectly, the results should be revised/corrected. Related to this, the experimental data in Figs. 2a and 3a show the same way symmetry breaking, too, whose physical origin should be explained/speculated. At the present stage, I am not convinced that the symmetry-breakings in experiments and numerics have the same origin. This is a rather serious question that requires an answer.

The neutron diffraction pattern collected in the experiment represent the structure factor of the material convoluted with the resolution function of the diffractometer. At the DMC diffractometer, the in-plane Q resolution is a lot better than the out-of-plane one, which effectively elongates the Bragg peaks in the $[\bar{1}10]$ direction (perpendicular to the scattering plane) and makes them look elliptic. When the angular profile of the intensity is then plotted (the intensity along the ring), the peaks appear broader with a lower amplitude or sharper with a higher amplitude (but preserving the area) depending on how the ellipse is oriented with respect to the ring. This may look as the peaks have somewhat different intensities on the color map. Because it is important to account for the ellipticity of the resolution function for accurate analysis, we prepared the dedicated sections in Supplementary Materials already in the previous version of the manuscript.

When fitting the data, we presciently accounted for the instrumental resolution. The extracted intrinsic peak broadening of the SSL was found identical for all six magnetic peaks. The fact that the peaks maintain the six-fold symmetry is also evident from our second experiment performed at the SANS-I instrument, where the resolution was more uniform (circular) [Figs. 1(g1,g2)] resulting in visually more symmetrical pattern on the colormap.

In our calculations, the shape of the peaks is also consistent with the six-fold symmetry. The figures showing the calculated structure factor present patterns that were convoluted with instrumental resolution for easier one-to-one comparison with the experimental data. In Fig. 2(b) these were convoluted with the DMC resolution function, and with the SANS-1 resolution function in Figs 4(b,c). An example of the calculated pattern before convolution (with peak shape preserving the six-fold symmetry) can be found in the Supplementary Materials in Figs. S4(a,b) for clarity. Because an individual run of the simulations may exhibit random preferred orientations of the propagation vector due to the limited system size, one can average the results of many runs that will be equivalent of the simulation of a larger system. However, in our simulations the peak intensities were quite similar because the system size of 300x300 spins was already large enough in most of the cases.